# ESTIMATING UNKNOWN POPULATION SIZES USING HYPERGEOMETRIC MAXIMUM LIKELIHOOD

## ABSTRACT

The multivariate hypergeometric distribution describes the fundamental process of sampling without replacement from a discrete population of elements divided into multiple categories. Despite the hypergeometric distribution's long history, the literature has not yet addressed the problem of maximum likelihood estimation when both the size of the total population and its constituent categories are unknown. Here, we show that this estimation challenge can be solved by maximizing the hypergeometric likelihood, even in the presence of severe under-sampling. We extend this approach to capture data generating processes where the ground-truth high-dimensional distribution is conditional on a continuous latent variable using the variational autoencoder framework, and validate the resulting model using simulated datasets. In a practical use case, we demonstrate that our method can recover the true number of gene transcripts present in a cell from sparse single-cell genomics data.

## 1 INTRODUCTION

The classic Pólya urn model (Eggenberger & Pólya, 1923) describes the process of randomly sampling from an urn containing balls of various colors, and is used to illustrate common discrete probability distributions that form the core building block of many probabilistic machine learning models. When balls are sampled from the urn *with* replacement, the distribution of counts of balls of each color is described by the multinomial distribution, whereas the hypergeometric distribution describes sampling *without* replacement. The hypergeometric distribution becomes important to successful modelling whenever the selection of one element from the distribution affects the probabilities of selecting subsequent elements (i.e. the counts are not independent), and when the sample size is significant compared to the population size. The hypergeometric distribution also enables the direct modeling of category counts, as opposed to category probabilities.

There are many common settings where it is valuable to model count data directly and to capture the dependence between category counts. For example, in the context of recommender systems and collaborative filtering, click data, song or movie play counts, and shopping basket item counts can all be thought of as being sampled without replacement. This is because the magnitude of counts is typically relatively small, and there is dependence between counts: watching a movie changes the probability of it being watched a second time, and adding an item to a virtual shopping basket changes the probability of it being added a second time.

In the field of single-cell genomics, the quantity of gene transcript count data measured at the resolution of individual cells is accumulating at an exponential rate. High-throughput experimental methods inherently sample without replacement, and with careful interpretation this single-cell data promises to answer important questions in biology and health. Due to the frequent occurrence of finite, discrete populations in nature, there are many other natural phenomena that can be modelled using the hypergeometric distribution.

Furthermore, the aforementioned applications are typically characterized by latent structure. Movie and music choices are driven by an underlying set of preferences, grocery purchases are driven by a person's diet and taste, and a cell's gene transcript counts depend strongly on its cell type. This low-rank structure induces correlation between features, such as songs of the same genre, and suggests that a latent-factor model is appropriate.

Despite the fundamental importance of the hypergeometric distribution, there is currently no effective way of estimating its parameters in many common settings of interest, including in the presence of high-dimensional data with intrinsic low-rank structure. In this paper, we describe an as yet unaddressed core problem for the hypergeometric distribution, namely performing maximum likelihood estimation when the number of elements in the overall population and in each constituent category are all unknown. We present a method for solving this problem, and show that the parameter estimation is tractable when there are two or more categories using empirical simulation. We then extend this approach using the variational autoencoder framework, enabling the estimation of high dimensional count data generated conditional on a latent variable. Finally, we demonstrate how this approach can be applied to recover the high number of missing gene transcript counts that are due to technical limitations of high-throughput single-cell genomics experimental methods.

## 2 BACKGROUND AND RELATED METHODS

### 2.1 THE HYPERGEOMETRIC DISTRIBUTION

Consider an urn that contains $N$ balls divided into two categories (colors): $N_1$ balls are white and $N_2 = N - N_1$ are black. Each ball has equal probability of being selected from the urn. If we sample $n < N$ balls without replacement, obtaining $c_1$ white counts and $c_2$ black counts, the distribution of counts of the number balls of each color that we obtain is given by the hypergeometric distribution, whose joint probability mass function is (Moivre, 1711):

$$P(c_1, c_2 | N_1, N_2) = \frac{\binom{N_1}{c_1}\binom{N_2}{c_2}}{\binom{N_1+N_2}{c_1+c_2}} \tag{1}$$

In general, when we have $K$ categories the joint probability mass function is:

$$P(c_1, \ldots, c_K | N_1, \ldots, N_K) = \frac{\prod_{i=1}^{K} \binom{N_i}{c_i}}{\binom{\sum_{i=1}^{K} N_i}{\sum_{i=1}^{K} c_i}} \tag{2}$$

The distribution for $K = 2$ is often called the univariate hypergeometric distribution, with $K > 2$ referred to as multivariate, but because in our problem setting there are already two unknown variables when $K = 2$, we use the name hypergeometric distribution for any $K \geq 2$.

### 2.2 EXISTING MAXIMUM LIKELIHOOD ESTIMATORS

There are two standard maximum likelihood estimation problems that have been investigated for the hypergeometric distribution.

*Known total population size:* When the total population size $N$ is known, the object is to estimate the true number of elements $N_i$ in each constituent category $i \in 1, \ldots, K$. The maximum likelihood estimator is then essentially the known population size scaled by the sample frequency of each category, with adjustment to ensure a correct integer solution (Oberhofer & Kaufmann, 1987).

*Unknown total population size*: In the more complex case, known as the capture-recapture problem (Darroch, 1958), the total population size is unknown. To estimate the total population, a sample is first drawn from the underlying distribution. All objects belonging to one of the categories in this sample are tagged, and all sampled elements are returned to the population. A second sample is then taken, and the number of tagged samples that reappear allow the estimation of the total population size using maximum likelihood. This method has found important applications in biology and ecology. However, it depends on the ability to tag and resample the same population.

Additionally, Tohma et al. (1991) propose a number of options for approximating the estimation of the parameters of the hypergeometric distribution when the category sizes are unknown, including an approximation using the likelihood, however they do not maximize the likelihood directly.

### 2.3 RELATED METHODS

The hypergeometric likelihood has for the most part been neglected in the modern machine learning context. Sutter et al. (2022) proposed a continuous relaxation of the non-central hypergeomet-

ric distribution to allow for differentiable sampling using the Gumbel-Softmax trick, and use it to learn category weights. This method assumes that the number of elements in each category are known, and therefore that only the category weights are to be estimated. Waudby-Smith & Ramdas (2020) presented a method for uncertainty quantification when sequentially sampling without replacement from a finite population, as defined by the hypergeometric distribution, to estimate confidence bounds as new data becomes available. Other parametric distributional assumptions can also be used to directly model count data, such as the Poisson and negative binomial, and a wide range of methods have been developed using these.

The multinomial distribution, which is the limiting form of the hypergeometric distribution when the sample size is negligible relative to the population ($N >> n$), appears frequently in machine learning literature. For example, it is often used to model counts indirectly by instead using the relative frequency of counts. LDA (Blei et al., 2003) uses a hierarchical generative framework to model the distribution of topics, documents over topics, and word (category) counts from a vocabulary (population) over documents. The distributions over topics and words are multinomial, with word counts being transformed into their relative frequency in the document. Liang et al. (2018) use a variational autoencoder with multinomial likelihood applied to collaborative filtering for recommender systems. The click data and play counts are binarized to accommodate the multinomial likelihood.

Awasthi et al. (2022) argue for the use of maximum likelihood estimation instead of empirical risk minimization, showing that it is better at capturing the appropriate inductive bias and that its performance is competitive with direct minimization of a target metric.

## 3 METHOD

Here, we assume that there exists a true discrete population with total number of elements $N$ divided into $K$ categories. We consider the setting where none of the category sizes are known, and we wish to estimate them from the data; that is, we do not know any category size $N_i$, nor the total population size $N = \sum_{i=1}^{K} N_i$. This scenario has yet to be addressed in the statistics literature, and we show that maximum likelihood estimation of all unknown population sizes is possible, even in the presence of severe under-sampling of the underlying distribution to be uncovered ($n << N$).

We consider $T$ independent trials, each producing an observed count vector $\boldsymbol{c}_t = \{c_{t,1}, \ldots, c_{t,K}\}$, $t \in 1, \ldots, T$. In trial $t$ we draw $n_t$ samples, without replacement, from the discrete underlying population, such that $\sum_{i=1}^{K} c_{t,i} = n_t$. The likelihood for the hypergeometric distribution is:

$$P(\boldsymbol{c}_1, \ldots, \boldsymbol{c}_T | N_1, \ldots, N_K) = \prod_{t=1}^{T} \frac{\prod_{i=1}^{K} \binom{N_i}{c_{t,i}}}{\binom{\sum_{i=1}^{K} N_i}{\sum_{i=1}^{K} c_{t,i}}} \tag{3}$$

The log-likelihood is:

$$\log P(\boldsymbol{c}_1, \ldots, \boldsymbol{c}_T | N_1, \ldots, N_K) = \sum_{t=1}^{T} \left[ \sum_{i=1}^{K} \log \binom{N_i}{c_{t,i}} - \log \binom{\sum_{i=1}^{K} N_{t,i}}{\sum_{i=1}^{K} c_{t,i}} \right] \tag{4}$$

The hypergeometric distribution is not part of the exponential family, so it is not clear that a closed-form maximum likelihood estimator exists, therefore we turn to numerical optimization methods. To enable continuous optimization, we consider a continuous and differentiable relaxation of the log-likelihood by replacing the factorials in the binomial coefficient with the gamma function, which is the extension of the factorial to real-numbered arguments.

$$\binom{a}{b} = \frac{a!}{b!(a-b)!} \tag{5}$$

$$= \frac{\Gamma(a+1)}{\Gamma(b+1)\Gamma(a-b+1)} \tag{6}$$

Using this log-likelihood for the parameter set $\theta = \{N_1, \ldots, N_K\}$, we perform maximum likelihood estimation obtain the MLE $\hat{\theta} \equiv \{\hat{N}_1, \ldots, \hat{N}_K\} = \arg\max_{\theta} \log p(\boldsymbol{c}_1, \ldots, \boldsymbol{c}_T | \theta)$.

The binomial coefficient is not defined if $c_k > \hat{N}_k$, which corresponds to the impossible scenario of sampling more balls of a given color than are present in the urn. To impose the requirement $N_k >= c_k$, we add a violation penalty $C_{\text{viol}}$ to the negative log-likelihood we seek to minimize, and we threshold any estimates of $\hat{N}_k < c_k$ at $c_k$ before evaluating the likelihood. We threshold at the observed sample value $c_i$ as opposed to the minimum across all samples $\min_i c_i$ to remain as general as possible. This is because in the case of multiple distributions we do not know which observation originates from which underlying distribution, and hence do not know what the correct minimum is.

$$C_{\text{viol}} = \sum_{i=1}^{K} \min(0, c_i - \hat{N}_i) \tag{7}$$

$$\hat{N}_i \leftarrow \max(c_i, \hat{N}_i) \quad i \in 1, \dots, K \tag{8}$$

In this paper we are interested in modelling scenarios where we have access to many samples drawn from the same underlying population, but where this population is under-sampled. Specifically, if $N$ is the total population size, we assume that we observe samples with at most $n_{max}$ objects drawn from the ground-truth distribution in each trial, giving a sample fraction $n_{max}/N$.

## 4 TRACTABILITY OF ESTIMATING UNKNOWN CATEGORY SIZES

Using empirical data simulation, we begin by demonstrating that this estimation problem is tractable in the ideal case where observations are sampled from a single discrete ground-truth distribution and the number of categories is small ($K = 2, 3$).

First, we select the true number of objects in each category, $N_1$ and $N_2$, with total $N = N_1 + N_2$. We then generate observations from $T$ trials, where in trial $t \in 1, \dots, T$ we sample $n_t$ objects from the true distribution, without replacement. For each trial, we first determine the number of objects to draw by sampling $n_t \sim \text{Uniform}(2, n_{max})$, where $n_{max} < N$ is the maximum number of objects that can be sampled in any trial. The max sample fraction is defined as $f_{max} = n_{max}/N$.

To demonstrate the typical behavior, we show results for a simulation for the scenario where $N_1 = 70$ and $N_2 = 30$ ($N = 100$, $K = 2$, $f_{max} = 0.4$), evaluating the negative log-likelihood (NLL) for all possible combinations of $\hat{N}_1$ and $\hat{N}_2$. Figure 1 shows the resulting loss landscape, where the minimum NLL value corresponds to the true $N_1$ and $N_2$ values. The lowest NLL region can be seen to elongated along the line of correct $N_2/N_1$ ratio. We note that the difference in NLL between the minimum and surrounding estimates is small, especially for estimates with the same ratio $N_1/N_2$.

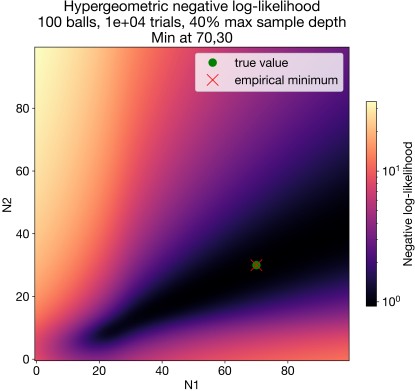

Figure 1: Negative log-likelihood landscape for $K = 2$ and $N = 100$.

We further investigate the accuracy of the maximum likelihood estimate for a wide range of numbers of observations $T$ and max sample fractions $f_{max}$. As the data is discrete, we use the Manhattan distance between the true and estimated counts to quantify the estimation error. In Figure 2a, we see that the location of the minimum NLL converges to the true value as the number of observations increases, for different levels of under-sampling. As expected, a higher max sample fraction results

in faster convergence and more accurate estimates, as quantified by lower error for the same number of samples. We extend this experiment to $K = 3$ (Figure 2b), where we see that the presence of more categories reduces the number of observations required to reach zero error. Here we use a confidence interval of 50% for visual clarity, and the same figure with a 90% confidence interval is included in Appendix B.

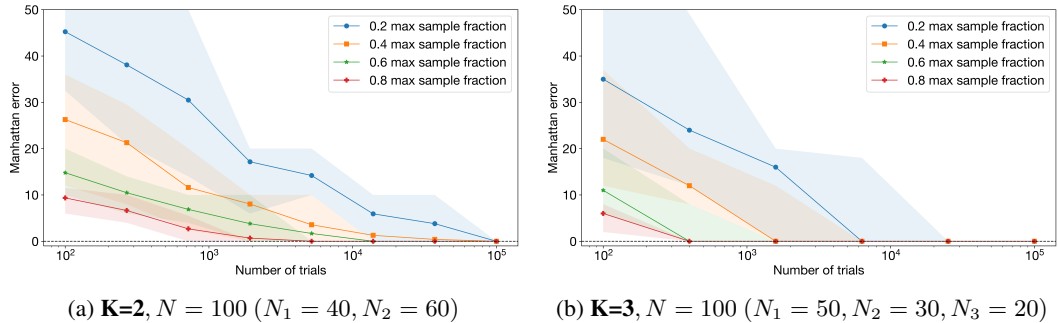

(a) **K=2**, $N = 100$ ($N_1 = 40$, $N_2 = 60$)   (b) **K=3**, $N = 100$ ($N_1 = 50$, $N_2 = 30$, $N_3 = 20$)

Figure 2: Maximum likelihood estimate Manhattan error for different numbers of observations at different max sample fractions ($N = 100$). 50% confidence interval over 50 random seeds.

Next, in Figure 3 we show that the maximum likelihood estimate can be obtained using gradient descent with the hypergeometric negative log-likelihood objective. We generate samples as before, and perform gradient descent using Adam with a learning rate of 0.1 and using a zero-initialization for the count parameters $\hat{N}_i$. We can see that the accuracy increases and the variance of the estimate decreases with increasing number of samples. We can also see that increasing the number of categories from two (Figure 3a) to three (Figure 3b) improves the rate of convergence of the estimate and reduces the final error for the same number of samples. This provides evidence that the bias of the maximum likelihood estimator decreases with increasing $K$. Additional experiments with different $f_{max}$ are included in Appendix B.

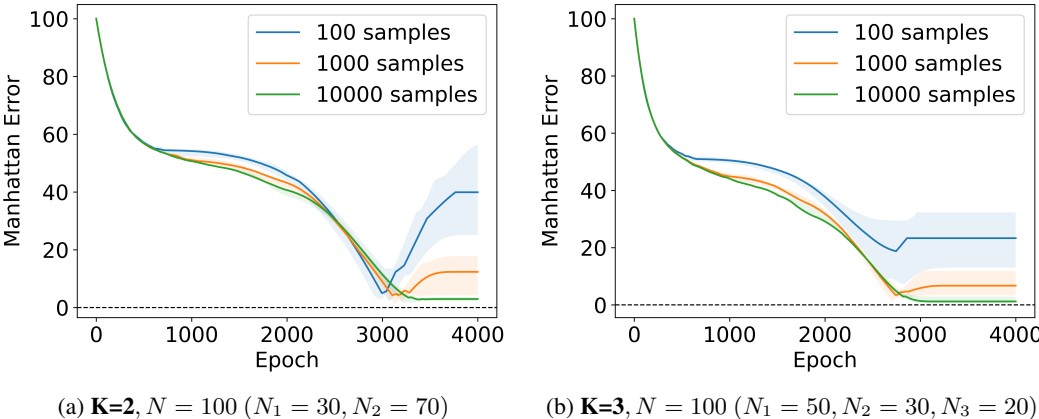

(a) **K=2**, $N = 100$ ($N_1 = 30$, $N_2 = 70$)   (b) **K=3**, $N = 100$ ($N_1 = 50$, $N_2 = 30$, $N_3 = 20$)

Figure 3: Maximum likelihood estimate Manhattan error per training epoch for different numbers of trials ($N = 100$, max sample fraction 0.4). The apparent increase in error following an initial decrease occurs because we measure absolute error, and this behavior corresponds to the estimate approaching and overshooting the true value. The estimates are more accurate when more trials are available, and exhibit a higher rate of convergence with larger $K$ (right). 50% confidence interval over 20 random seeds.

## 5 VARIATIONAL AUTOENCODER WITH HYPERGEOMETRIC LIKELIHOOD

We extend our method to allow for the estimation of a high-dimensional distribution that is conditional on a continuous latent variable, allowing it to capture a continuous mixture of count distri-

butions. This modelling assumption is essential when the true distribution is considerably under-sampled, leading to sparsity, and so information needs to be shared between similar observations to successfully model the data. For example, collaborative filtering for movie recommendation was one successful approach to the Netflix Prize. Movie recommendation is challenging because each individual provides ratings for few movies, but this can be overcome by leveraging patterns across many individuals with similar tastes. The variational autoencoder (VAE) (Kingma & Welling, 2014) is a powerful framework for performing efficient estimation of the generative distribution parameters in the presence of a continuous latent variable and big data.

Following the VAE framework, we assume a data generating process where a latent variable $z$ is first drawn from a prior distribution $p(z)$, then a count vector $c$ is generated from the conditional hypergeometric likelihood $p(c|z)$. Note again that we are using the continuous relaxation of the hypergeometric distribution, so the generated $c$ are continuous. The marginal hypergeometric likelihood that we are interested in, $p(c) = \int p(c|z)p(z)dz$, is intractable. We approximate the true unknown posterior for the latent variable $p(z|c)$ with the variational distribution $q_\phi(z|c)$, parameterized by a deep feed-forward neural network with parameters $\phi$. Similarly, we represent the parameters of conditional hypergeometric likelihood $p_\theta(c|z)$ with another neural network with parameters $\theta$. We choose a factorized multivariate Gaussian as the prior $p(z)$ over the latent variable. We optimize the variational lower bound augmented by the violation penalty:

$$\mathcal{L}(\theta, \phi; c_t) = -D_{\text{KL}}(q_\phi(z|c_t)||p(z)) + \log p_\theta(c_t|z) + \sum_{i=1}^{K} \max(0, c_i - \hat{N}_i) \quad (9)$$

The first term is the KL divergence between the true and approximate posterior, the second term is the log-likelihood, and the third term is the violation penalty.

To generate a simulated dataset, we first specify $M$ different count distributions over $K \geq 2$ features (categories). We sample an equal number of count vectors $c$ from each count distribution, where the number of counts per trial is drawn independently from $n_t \sim \text{Uniform}(n_{min}, n_{max})$.

We specifically simulate two ground-truth distributions over $K = 1000$ categories, where one distribution is composed of $10,000$ total counts and the other $30,000$. These distributions are generated from two randomly sampled probability vectors of length $K$, which we then multiply by the total desired number of counts. Finally, we round each count down to the nearest integer to obtain a discrete distribution. We simulate $100,000$ total observations, where for each we sample $n_t \sim \text{Uniform}(100, 5000)$ counts, resulting in observations with at most 60% and 20% of the total number of elements for the two distributions, respectively. We implement early stopping by ending the model training when the validation negative log-likelihood enters a plateau.

Figure 4 shows the trajectory of count estimates over the course of training, demonstrating convergence of the estimates to their correct total populations. We emphasize that despite the model not known the true number of distributions, and therefore not having access to the labels of the observations, it is correctly able to learn a latent space that perfectly separates the two sets of samples. Note that both distributions are sampled to the same $n_{max}$, so this disinction is not due simply to differences in total observed counts for each distribution

Figure 5 compares the distribution of observed and estimated counts, with 5a showing that our method is able to recover the original population sizes despite high sparsity (false zero counts) and under-sampling. For the two ground-truth distributions with different $N$, the estimates are shifted away from the observed distribution and approach the true values. We see that the variance of the estimate is higher when the under-sampling is more drastic (20% vs 60%). Figure 5b shows the original and estimated count distributions for the top three categories by mean ground-truth count, again showing that the estimates approach the ground-truth values.

## 6 APPLICATION TO SINGLE-CELL GENOMICS

The detailed measurement of the contents of individual cells promises to vastly improve our understanding of fundamental biology. In recent years, the advancement of high-throughput techniques in the field of single-cell genomics has enabled the collection of large numbers of gene transcripts from individual cells, resulting in vast count matrices. Each cell has a finite population of transcripts that can be captured - on the order of one hundred thousand to a million - and a transcript can only

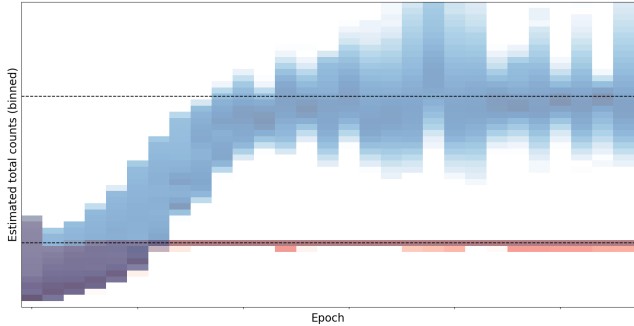

Figure 4: Vertical binning of the estimated total counts per observation vs training epochs. Black lines are true total counts for the two distributions. The final estimate recovers the ground-truth population sizes for both distributions.

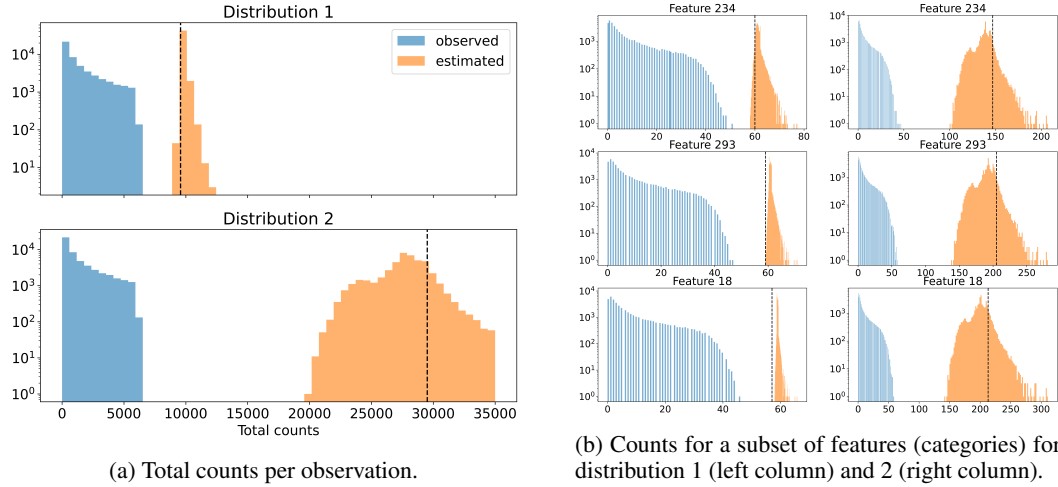

(a) Total counts per observation.

(b) Counts for a subset of features (categories) for distribution 1 (left column) and 2 (right column).

Figure 5: Histograms comparing the observed (blue) and estimated (orange) counts to the ground-truth underlying distribution. Our model estimate shifts the count distributions away from zero and close to ground-truth value (dashed line).

be captured once prior to sequencing. It is accepted that the main source of technical noise in this data is due to under-sampling (Kuo et al., 2023), leading to the well-known phenomenon of dropout (inflated occurrence of zeros in the final count matrix). This technical noise hinders our ability to draw meaningful scientific conclusions from these experiments.

The hypergeometric distribution is well-suited to modelling this capture process, as the number of captured gene transcripts in an experiment is significant relative to the total population size, and capture occurs without replacement, leading to dependence between gene feature counts. The high-dimensional distribution of gene transcript counts can be effectively represented by latent variable models Lopez et al. (2018), as individual genes are often members of co-expression networks which result in highly correlated counts. We therefore use our VAE approach with the hypergeometric likelihood, with genes as categories, in order to recover the true gene transcripts counts in each cell from the sparse, under-sampled count data.

Because there is typically no way of knowing the true number of transcripts of each gene in a given cell, we focus on what is known as a spike-in experiment, which does provide a ground-truth. In the experiment we consider (Ziegenhain et al., 2022), a known concentration of a solution of synthetic RNA is placed in small wells, and human cells are individually placed in a subset of the wells. After the transcripts present in these wells are captured and sequenced, we obtain experimental counts corresponding to both the human RNA (for which the original amount is unknown) and the synthetic RNA (original amount is known). The wells that did not have cells in them should have

an equal amount of the synthetic RNA across measurements, so we can therefore use the measured counts of synthetic RNA in empty and cell-containing wells as a ground-truth reference to evaluate our estimated human gene transcript counts.

*Dataset*: The **SPIKE** dataset consists of counts for 43k genes (categories) across 1126 observations. Of these, 181 observations contain a mixture of synthetic RNA and human kidney cells (labelled "HEK293T"), and the remainder 945 contain only the synthetic RNA (labelled "empty").

Figure 6 compares the distribution of measured counts in this dataset for one specific synthetic RNA (#12) in the empty and HEK293T measurements. This exemplifies the stochastic under-sampling that occurs, as we expect the true counts should be identical across all samples (approximately 4000 for this particular spike-in RNA). It is also clear that the presence of human RNA significantly lowers the amount of synthetic RNA that is captured, which supports the assumption that there is dependence between counts, i.e. a transcript is captured at the cost of another.

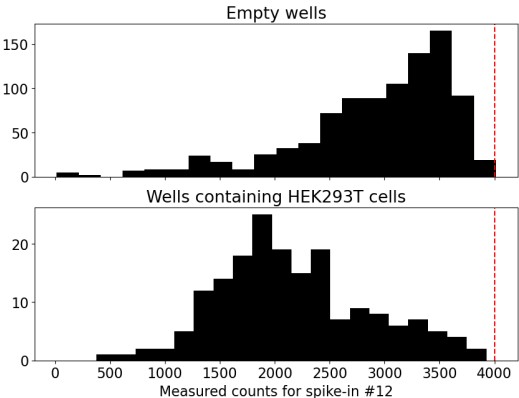

Figure 6: Measured counts for synthetic spike-in RNA #12 with and without human cells present. Red dashed line is ground-truth count (ideally all measurements should be equal).

We train our VAE model with hypergeometric likelihood on the **SPIKE** dataset to recover the true count matrix. We use the top 10,000 genes (categories) by mean transcript count, for computational efficiency and because many genes are not expressed in this celltype. Model and training hyperparameters are given in Appendix A. Summary results of the final estimated count matrix are shown in 7.

Figure 7a shows the maximum likelihood estimate for the total number of counts per observation (the sum of the estimated count matrix rows). The authors estimate the true total amount of synthetic RNA per observation as approximately 30,000, which closely aligns with our model estimate. The unknown total amount of RNA in the human kidney cells is estimated by our model to have a median of 260,000. Note that the number of observations available in this dataset is very small compared to typical single-cell datasets (without ground-truth), and this estimate would likely be even more accurate and have a lower variance with a larger number of observed cells. 7b shows the estimated counts specifically for spike-in #12, whose ground-truth value is approximately 4000. We can see that although the initial distribution of this synthetic RNA's counts have a significantly different mean in the empty and cell-containing observations, our model produces an estimate near the true value, and brings both distributions into alignment. These results are similar across other spike-in RNA. These results show that we are to recover the known ground-truth distribution of synthetic RNA that has been corrupted by the stochastic capture process, and this allows us to learn new information about the unknown distribution of RNA counts in the human kidney cells.

## 7    CONCLUSION

We propose a method for estimating the unknown category sizes of a discrete distribution of elements that is sampled without replacement, using a continuous relaxation of the hypergeometric likelihood. Through empirical data simulation we show that the maximum likelihood estimate is

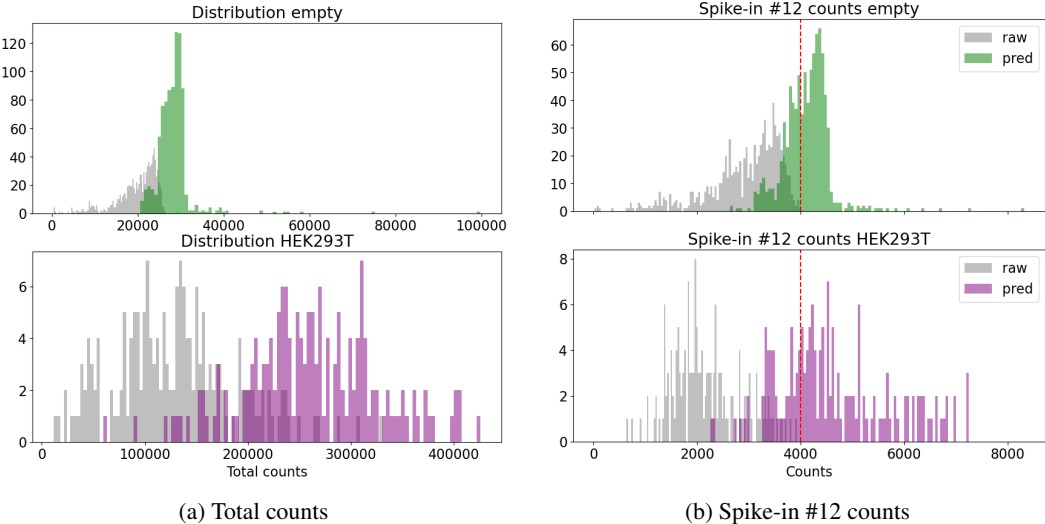

(a) Total counts

(b) Spike-in #12 counts

Figure 7: Histograms of the number of measured counts when only synthetic RNA is present (top) and when both synthetic RNA and human RNA (from kidney cells) are present (bottom). The original (measured) distribution is in grey, and the red dashed line shows the ground-truth amount of synthetic RNA #12.

accurate with sufficient observations, over a range of maximum sample fractions. We show that our approach can be extended to model a distribution that is conditional on a continuous latent variable using the variational autoencoder framework. We address gene transcript count sparsity, an emerging obstacle in the field of single-cell genomics, and show that our method is able to recover the true number of transcripts in a cell. Due to the prevalence of finitely sampled discrete populations in biology and beyond, we expect this method can be successfully used in many other application domains.

## 8 FUTURE WORK

In this work we demonstrated the value of our approach using single-cell genomics data, however there are a number of other applications that could likely benefit from the same approach. In particular, we expect collaborative filtering applications for music, movies, and shopping cart data to be promising applications. However, the evaluation metric will need to be modified in the absence of a direct ground-truth for comparison. While the violation penalty we presented is simple and effective, other methods such as bounded optimzation may result in faster convergence. Although we performed experiments using a variety of $f_{max}$, it would be valuable to gain a better understanding of how to detect if $f_{max}$ (max counts sampled vs total number of counts) is too low for a new dataset, particularly if we do not have good prior knowledge of the expected magnitude of counts. Finally, while our empirical experiments have shown promising results, we hope future research will develop a better theoretical understanding of the log-likelihood for the continuous relaxation of the hypergeometric distribution.

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

## A HYPERPARAMETERS

Table 1: Experiment hyperparameters

| Parameter | Simulated | SPIKE |
|---|---|---|
| # features | 1000 | 10,000 |
| Encoder layers | 64, 64 | 128, 128 |
| Decoder layers | 128, 128 | 256, 256 |
| Latent space dimension | 4 | 16 |
| Learning rate | 0.01 | 0.01 |
| Batch size | 500 | 563 |
| Violation penalty min/max | 5,50 | 1,100 |

## B ADDITIONAL FIGURES

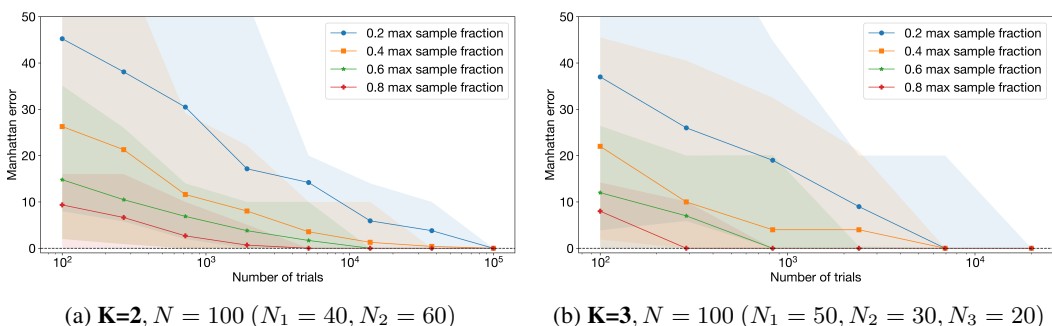

(a) **K=2**, $N = 100$ ($N_1 = 40, N_2 = 60$)  (b) **K=3**, $N = 100$ ($N_1 = 50, N_2 = 30, N_3 = 20$)

Figure 8: Maximum likelihood estimate Manhattan error for different numbers of observations at different max sample fractions ($N = 100$). 90% confidence interval over 50 random seeds.

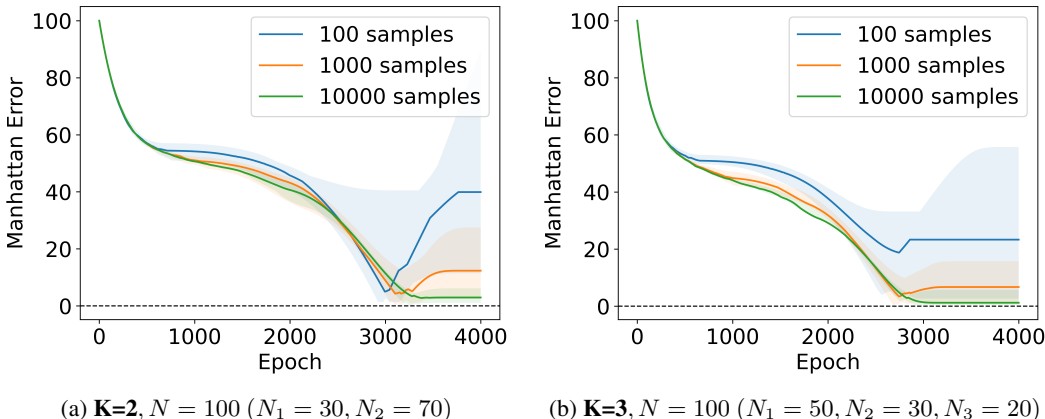

(a) **K=2**, $N = 100$ ($N_1 = 30, N_2 = 70$)  (b) **K=3**, $N = 100$ ($N_1 = 50, N_2 = 30, N_3 = 20$)

Figure 9: Maximum likelihood estimate Manhattan error per training epoch for different numbers of trials ($N = 100$, max sample fraction 0.4). 90% confidence interval over 20 random seeds.

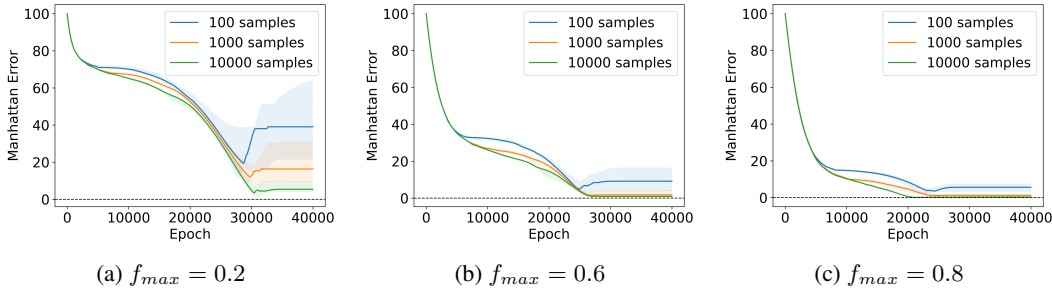

(a) $f_{max} = 0.2$        (b) $f_{max} = 0.6$        (c) $f_{max} = 0.8$

Figure 10: Maximum likelihood estimate Manhattan error per training epoch for different numbers of trials and different $f_{max}$ ($K = 3, N = 100, N_1 = 50, N_2 = 30, N_3 = 20$). 50% confidence interval over 20 random seeds.

