# OpenReview forum: "Estimating Unknown Population Sizes Using Hypergeometric Maximum Likelihood"
_ICLR.cc/2024/Conference — Submitted to ICLR 2024_

### Official Review · Reviewer_etSf · 2023-10-30

**Soundness:** 2 fair
**Presentation:** 4 excellent
**Contribution:** 2 fair
**Rating:** 5
**Confidence:** 4

**Summary:**

This paper considers the problem of estimating the population counts in a multivariate hypergeometric distribution.  The authors first propose using a continuous function that agrees with the likelihood function of the hypergeometric distribution on integer values, and then they maximize this surrogate function.  They show that this approach works if one has access to several independent samples (which would correspond to drawing a sample without replacement, then replacing all of those draws, and then sampling without replacement again, and so on).  The authors then turn to a more general case where different observations are each a set of samples drawn without replacement, and each observation comes from a different population (so that each observation is a hypergeometric distributed random variable with a different distribution).  They propose using a variational autoencoder in this case to perform inference, and apply this approach to single cell RNA sequencing data.

**Strengths:**

* The paper is generally quite clear and the writing is good.
* In principle, I like the idea of considering UMIs (i.e., individual transcripts) in single cell datasets as forming a population and so datasets are hypergeometric draws.  This is conceptually quite clean, and a nice extension of the normal model to take the weak dependency between observations into account.  I think this problem is well-appreciated in the field (e.g., treating data compositionally induces similar effects), but as far as I know this is a new approach to the problem. This point is made particularly well in Figure 6.

**Weaknesses:**

* I have some concerns about the motivation/applicability of the paper. In the first part, the authors contrast their approach to capture-recapture, and say that capture re-capture is not good because one must be able to sample (to tag) and then resample from the same population.  Yet, in their own setup (e.g., Equation (3)) the authors assume that one can repeatedly and independently sample from the same population.  I find this setup interesting from a statistics perspective, but unrealistic as an experimental design.  If I understand correctly, it would correspond to sampling without replacement from a population, returning all of those samples to the population, and then resampling without replacement and so on.  In some sense, this is essentially what the authors claimed was problematic about capture-recapture, but they are repeating it $T$ times instead of twice.  I would be happy to change my mind about this being problematic if the authors could provide some compelling, real life examples of datasets of practical interest with this experimental design.
* At the bottom of Figure 3 it is claimed that Equation (4) is concave.  I plotted Equation (4) for a particular setup and it is _not_ concave.  One can also see this from Figure 1 where if one were to draw a well place diagonal line through the plot, one could obtain a linear slice where the negative log-likelihood first decreases, then increases, then decreases again, which is impossible for a convex function.  The error in reasoning comes from the fact that Equation (4) is the sum of both $\log \Gamma(x)$ and $-\log \Gamma(x)$ functions.  The former is strictly concave, but the latter is strictly convex.  The sum of convex and concave functions does not need to be either convex or concave.
* Figures 2 and 3 should contain error bars and/or multiple replicates.  The fact that the error fluctuates across the number of trials (as opposed to monotonically decreasing) suggests that there is some substantial noise.  This noise makes it very difficult to assess the claim that increasing $K$ makes the problem easier.
* My intuition is that the good performance in Figures 2 and 3 comes from the large number of trials.  E.g., even at thousands of trials, there is still substantial error.  In particular, if one only has a single trial, then I think that the MLE for the population counts should just be the observed counts, but the magnitude of this is driven entirely by the sample size and has nothing to do with the population size.  As a result, I don't think this is a sensible approach when there is only a single trial.  I suspect that this becomes problematic in the VAE setting -- since the likelihood is not particularly informative for any given point the prior will be incredibly important.


Minor points / typos:
* In Equation (1), I believe that $n_2$ should be $c_2$
* The notation in Equation (8) is a bit sloppy.  I understand the point the authors are making, but it seems sloppy to define $\hat{N}_i$ as a function of itself (i.e., $\hat{N}_i$ appears on both sides of the definition).
* "to elongated along the line of correct" --> "as a line along the correct"

**Questions:**

* I believe that the continuous version of the likelihood (i.e., Equation (4) but using equation (6) for the binomial coefficients) does not define a proper probability distribution for non-integer $N_k$.  Is this problematic?
* Is there a reason for including the violation penalty $C_\text{viol}$ as opposed t just performing bounded optimization?  The lower bound on $\hat{N}_i$ can be determined easily from the data, so it would not be difficult to enforce the bound durning optimization.  Furthermore, the constraint set is quite simple (and convex), so something like projected gradient descent would be easy to perform.

---

> ### Author Response · Authors · 2023-11-18
> **Author comment (1/2)**
>
> Thank you for your review! We have addressed your questions and comments below, and uploaded a new revision of the paper, implementing your suggestions and updating the figures.
>
> > I have some concerns about the motivation/applicability of the paper. In the first part, the authors contrast their approach to capture-recapture, and say that capture re-capture is not good because one must be able to sample (to tag) and then resample from the same population. Yet, in their own setup (e.g., Equation (3)) the authors assume that one can repeatedly and independently sample from the same population. I find this setup interesting from a statistics perspective, but unrealistic as an experimental design. If I understand correctly, it would correspond to sampling without replacement from a population, returning all of those samples to the population, and then resampling without replacement and so on. In some sense, this is essentially what the authors claimed was problematic about capture-recapture, but they are repeating it N times instead of twice. I would be happy to change my mind about this being problematic if the authors could provide some compelling, real life examples of datasets of practical interest with this experimental design.
>
> One example of a real-life application that directly matches this experimental design is a variation of the standard capture-recapture problem of counting fish in a lake. In the original problem statement, we wish to estimate the number of fish of a certain species in a lake. To do so, we catch a sample of fish, tag the fish of the species of interest, and return all caught fish to the lake. We then catch a second sample and count how many tagged fish have been recaptured. However, consider a variation where we are instead counting fish in a flowing river. Here it would be impossible to recapture any tagged fish since they are swept downriver. But if we can assume that the instantaneous distribution of fish at the sample point is relatively constant over the course of the experiment, then we can perform multiple resamplings, counting the fish of the species of interest each time, and obtain an estimate using our proposed method.
>
> As a second hypothetical scenario, in the context of quality control we may wish to estimate the number of defects in a batched production line, where it is known that the same machines are used to produce each batch. Capture-recapture would require sampling a subset of items from a batch, marking those with defects, returning them, sampling another random subset from the same batch, and counting how many of the marked defective items reappear. This would require stopping the production line, or removing the entire batch until the experiment is complete. In contrast, here we propose sampling from each batch only once, without needing to mark and resample items twice from the same batch, allowing the production line to continue operating.
>
> We believe that the key difference between the capture-recapture method and our repeated sampling approach is that the former requires resampling of the exact same population, whereas we only require resampling from the same distribution. Therefore in any experiment where we can assume the distribution remains unchanged across multiple instances, it would be possible to resample the same distribution without resampling the exact same population.
>
> Additionally, capture-recapture cannot be used if carrying out the experiment changes the underlying distribution, for example if the act of sampling from a batch may cause additional defects. In our later experiment this is important as measuring gene transcripts requires destroying the cell they are in.
>
> Having said that, we agree that the classical statistics regime, where there is only a single count distribution to be estimated, is not the most valuable setting to apply our approach. We mainly intend this initial set of simulations to demonstrate that it is theoretically possible to perform inference on the multivariate hypergeometric distribution with unknown population sizes, as we do not believe this has been demonstrated previously. We intend to set the scene for the later sections, and we believe that the primary method of practical value is in the case of count distributions conditional on a continuous latent variable. It would be impossible to apply capture-recapture to the single-cell genomics problem without substantial breakthroughs on the experimental side.

---

> > ### Comment · Reviewer_etSf · 2023-11-21
> >
> > Thank you for your comments.  I am still not convinced about the motivation.  It feels to me that assuming the "instantaneous distribution of fish at the sample point is relatively constant" is a bigger deviation from reality than assuming that we are sampling with replacement (or sampling without replacement from an infinite population), but of course that is somewhat subjective.
> >
> > In general, I find it implausible that one can "resample the same distribution without resampling the exact same population".  In this setting "the same distribution" means the exact same numbers of each type, and it just seems unrealistic to assume that two different populations would have exactly the same numbers of each type.
> >
> > I do appreciate the work put into the other responses, and as a result I have slightly raised my score.

---

> > > ### Author Response · Authors · 2023-11-22
> > >
> > > We could have presented our main approach (Section 5) directly; however, for clarity we chose to first demonstrate that inference using the hypergeometric likelihood was tractable in the single distribution case. Would you find the motivation more convincing if we integrated Section 5 (VAE) into Section 3 (Method)?
> > >
> > > We would like to re-emphasize that it is not necessary for the initial simple distribution case (Section 3) to have a perfect experimental application in order for our main approach (Section 5) to work. This initial case is simply a mathematical stepping stone towards the main approach applied in Sections 5 and 6, and our conclusions hold without it. In later sections, sampling the same conditional distribution (the distribution of RNA in cells of the same type) without resampling the same population (the same cell, since it is destroyed) is a justifiable and necessary modeling assumption.

---

> ### Author Response · Authors · 2023-11-18
> **Author comment (2/2)**
>
> > At the bottom of Figure 3 it is claimed that Equation (4) is concave. I plotted Equation (4) for a particular setup and it is not concave. One can also see this from Figure 1 where if one were to draw a well placed diagonal line through the plot, one could obtain a linear slice where the negative log-likelihood first decreases, then increases, then decreases again, which is impossible for a convex function. The error in reasoning comes from the fact that Equation (4) is the sum of both X and X functions. The former is strictly concave, but the latter is strictly convex. The sum of convex and concave functions does not need to be either convex or concave.
>
> Thank you for pointing this out, we agree with you and so have removed this statement. Please note that the relevance of our results and conclusions remains intact. While the log-likelihood is not concave, it appears to be concave along the line of correct N_1/N_2 ratio, however have not yet been able to show this rigorously. This is important as the optimization path is mainly along this vector. We hope that deeper theoretical analysis of his likelihood will be undertaken in the future, and have added this to the Future Work section.
> > Figures 2 and 3 should contain error bars and/or multiple replicates. The fact that the error fluctuates across the number of trials (as opposed to monotonically decreasing) suggests that there is some substantial noise. This noise makes it very difficult to assess the claim that increasing X makes the problem easier.
>
> We are grateful for this suggestion. We have added confidence intervals to Figures 2 and 3 by repeating each trial with multiple random seeds. We’ve used 25/75% intervals for visual clarity, and the same figure with 5/95% intervals has been added to the appendix.
>
> > My intuition is that the good performance in Figures 2 and 3 comes from the large number of trials. E.g., even at thousands of trials, there is still substantial error. In particular, if one only has a single trial, then I think that the MLE for the population counts should just be the observed counts, but the magnitude of this is driven entirely by the sample size and has nothing to do with the population size. As a result, I don't think this is a sensible approach when there is only a single trial. I suspect that this becomes problematic in the VAE setting -- since the likelihood is not particularly informative for any given point the prior will be incredibly important.
>
> We agree that this approach only works when there are many observations available, and not in the case of few trials. In our experiments we have observed that with few observations the MLE obtains the correct relative fraction of counts (as would be obtained using a multinomial likelihood), but that more observations are required to obtain the exact counts. The log-likelihood is quite flat, and therefore the minimum is sensitive when there are few observations.
>
> We have not investigated if the estimate converges more quickly if a prior is used, and have focused on applications where there are enough trials to obtain a good estimate. It would be an interesting extension to see if fewer trials are required with a well-chosen prior.
>
> > In Equation (1), I believe that n_2 should be c_2.
>
> Yes you are correct, we have fixed the typo.
>
> > The notation in Equation (8) is a bit sloppy. I understand the point the authors are making, but it seems sloppy to define N_i as a function of itself (i.e., N_i appears on both sides of the definition).
>
> That is a good point, we have changed the equation from an equality to an update.
>
> > I believe that the continuous version of the likelihood (i.e., Equation (4) but using equation (6) for the binomial coefficients) does not define a proper probability distribution for non-integer N_k. Is this problematic?
>
> You are correct, the continuous relaxation is no longer properly normalized for non-integer N_k. We do not believe this is a problem if we are using it as an optimization objective, as the parameters that produce the minimum remain unchanged.
> > Is there a reason for including the violation penalty C_k as opposed to just performing bounded optimization? The lower bound on N_i can be determined easily from the data, so it would not be difficult to enforce the bound during optimization. Furthermore, the constraint set is quite simple (and convex), so something like projected gradient descent would be easy to perform.
>
> We did not pursue a bounded optimization approach as we were not able to obtain a lower bound on N_i when there are multiple distributions to estimate. We are in the unsupervised regime and do not know which observations belong to which distribution, and hence what lower bound is appropriate for each observation. However, we expect that there is room for more sophisticated ways of discouraging count estimate violations, and we have added bounded optimization as a possible next step to consider.

---

### Official Review · Reviewer_rSc3 · 2023-11-01

**Soundness:** 3 good
**Presentation:** 3 good
**Contribution:** 2 fair
**Rating:** 5
**Confidence:** 3

**Summary:**

The paper introduces a method for estimating category sizes of a multivariate hypergeometric distribution, which models the process of sampling without replacement from a discrete population with multiple categories. The authors employ a continuous and differentiable relaxation of the hypergeometric likelihood using the gamma function to replace factorials in the binomial coefficients. Through empirical data simulations, they demonstrate the accurate recovery of category sizes, particularly in scenarios with a single ground-truth distribution and a limited number of categories.

Furthermore, the authors expand their approach to model a data generation process that incorporates a latent variable 'z' within the framework of a variational autoencoder. To show the effectiveness of their method, they present results from both simulated and real data experiments.

**Strengths:**

The paper tackles an interesting problem with significant applications in the field of biology. It is clearly written, offering readers the necessary background to facilitate their understanding.

**Weaknesses:**

- The authors do not provide a clear motivation for the second part of the work which focuses on the variational autoencoder framework. Although the initial experiments focus on the recovery of the category sizes of a single multivariate hypergeometric distribution, they later explore a scenario where they have a mixture of distributions. The estimated counts are per observation, what is the ultimate goal in this case?

- Given that the authors rely on only on empirical simulations to support the effectiveness of their method I would expect more thorough and convincing experimental results.

**Questions:**

Q1: What do you mean by low-rank structure in this part: "including in the presence of high-dimensional data with intrinsic low-rank structure." ?

Q2: It seems to be a mistake in the likelihood of equations 3 and 4 compared to that at the end of page 3. What is the optimization problem that you solve?

Q3: In the experiment at section 5, the model does use the knowledge of the number of distributions, which is 2. How do you estimate the total counts of the two distributions? What happens in the case that you have more than 2?

Minor: N2 should be 30 in section 4.

---

> ### Author Response · Authors · 2023-11-18
> **Author comment**
>
> Thank you for your review! We have addressed your questions and comments below, and uploaded a new revision of the paper, implementing your suggestions and updating the figures.
>
> > The authors do not provide a clear motivation for the second part of the work which focuses on the variational autoencoder framework. Although the initial experiments focus on the recovery of the category sizes of a single multivariate hypergeometric distribution, they later explore a scenario where they have a mixture of distributions. The estimated counts are per observation, what is the ultimate goal in this case?
>
> Thank you for this feedback, we have added additional description of the motivation for the need to model a mixture of distributions to the introduction and Section 5.
>
> In machine learning, count data is often binarized so that it can be modeled using a probability distribution over categories, such as by using a neural network with a softmax output layer. However, we believe that disregarding the actual counts leads to loss of information that may be valuable in certain applications. Therefore, departing from previous work, the ultimate goal of our approach is to directly estimate the observed counts without forced binarization, thus preserving this information.
>
> > What do you mean by low-rank structure in this part: "including in the presence of high-dimensional data with intrinsic low-rank structure." ?
>
> Here we are specifically referring to data that can be described by a small set of latent patterns across features without much loss of information, classically recovered by using PCA or now autoencoder neural networks. An intuitive example is that an individual’s song play count data can substantially be estimated using a minimal description of their favorite music genres. We have added clarification in the introduction. In this paper we choose to use a variational autoencoder, which inherently learns to maximize the likelihood of the reconstructed data based on an information bottleneck represented by a learned latent space. So the low-rank structure here would be captured by a latent space that is of much smaller dimension than the number of input data dimensions. The embedding dimension is treated as a hyperparameter.
>
> > It seems to be a mistake in the likelihood of equations 3 and 4 compared to that at the end of page 3. What is the optimization problem that you solve?
>
> Thank you for pointing this out, we have updated the notation to make it clear which parameters are being optimized, and to distinguish between the parameters and the observations.
>
> > In the experiment at section 5, the model does use the knowledge of the number of distributions, which is 2. How do you estimate the total counts of the two distributions? What happens in the case that you have more than 2?
>
> To clarify, the VAE models that we train in Sections 5 and 6 does not have access to the true number of distributions present in the training data. Unlike a classical latent-factor model like a Gaussian Mixture Model, the number of categories is learned implicitly in the latent space of the VAE, so as to maximize the log-likelihood. The fact that the number of distributions is not available to the model may not have been clear because we visualized the results using the true distribution labels, which allows us to show that the model does learn the two distributions correctly. We have added language to emphasize this important fact.
>
> The model estimates the total number of counts per feature for each observation. We then sum across all features for each observation to get the estimated total counts - this is the number that is shown in Figure 4, 5a, and 7a.
>
> In the case of larger N, as long as the latent space is sufficiently expressive there should be no problem with capturing additional distributions. It is common to use a VAE to perform unsupervised decomposition in the latent space.
>
> > N2 should be 30 in section 4.
>
> You are correct, we have fixed that typo - thank you.

---

### Official Review · Reviewer_XXPF · 2023-11-06

**Soundness:** 3 good
**Presentation:** 3 good
**Contribution:** 3 good
**Rating:** 6
**Confidence:** 4

**Summary:**

This paper considers maximum likelihood estimation in the multivariate hypergeometric model given that both the total population and the number of elements in each category are unknown. After a brief review of related methods for estimation in simpler scenarios, or when resampling is possible (i.e., capture-recapture), the main method is presented. The key idea is to use a relaxation of a standard hypergeometric likelihood function (by replacing factorials with the Gamma function) with appropriate constraints to ensure that the estimated total population sizes are not greater than the sampled counts. Some experiments on synthetic data are done to check the proposed estimation procedure as it depends on the maximum samples drawn and the number of underlying categories. Then, a larger scale experiment with a VAE, using synthetic and real-data from single cell genomics, is presented. It is shown that the proposed method works well in recovering the underlying counts when the ground truth is known (in both synthetic and real-data scenarios).

**Strengths:**

This paper is very clearly written, easy to read, and makes the contribution clear. The application is well-chosen and the experiments are illustrative. It is a bit surprising that something that like what the authors have proposed has not been done before, but this is perhaps due to the relatively rarer use of relaxation strategies (namely, replacing discrete with continuous variables) in the statistics literature.

**Weaknesses:**

I would have liked to seen more examples of potential applications in the machine learning context. The authors mention recommender systems as a possible application, as well as applications in language. However this is not commented on or developed further. What about applications in a statistical context? The authors should clearly describe some other potential contexts where we can have multiple independent samples without replacement where this model would be applicable. I believe the method is also applicable to the non-central Hypergeometric as in the cited Sutter et al. (2020) paper and it would have been interesting to apply the proposed method to scenarios that are similar to those described in that work.

**Questions:**

It feels like there is a bit of a gap in the experiments, there is a jump from using 3 categories to more than 1000 categories. What about more intermediate cases, say with 100 categories? How does the proposed method work in this case with respect to n_max as well as the number of samples?

Has a similar relaxation (replacing a Binomial coefficient or factorials with a Gamma function) been used elsewhere that you are aware of in estimation contexts?

There is a typo in equation (1), you should have c_2 and not n_2?

In equation (7), shouldn't you define c_i to be the largest category count over the T samples, as that gives you the lower bound on the N_k?

---

> ### Author Response · Authors · 2023-11-18
> **Author comment (1/2)**
>
> Thank you for your review! We have addressed your questions and comments below, and uploaded a new revision of the paper, implementing your suggestions and updating the figures.
>
> > I would have liked to see more examples for potential application in machine learning (develop proposed examples further) or statistical context
>
> We agree that the more typical machine learning examples we proposed remain to be investigated in more detail, and we hope that this paper will encourage others to build on this work. Part of the reason for choosing to apply our method to single-cell genomics data was the availability of a ground-truth to which to compare our estimate, which we were not able to find for the other applications we mention. We now acknowledge this aspect in the Future Work section of the improved paper draft.
>
> The most promising application that is similar to those commonly addressed in machine learning is that of collaborative filtering for user preference count data (songs, movies, online marketplace purchases). For example, in the highly cited paper “Variational Autoencoders for Collaborative Filtering” the main contribution is to model the user-by-item click (i.e. count) matrix using a VAE to estimate the parameters of a multinomial likelihood. These authors state that “for simplicity, we binarize the click matrix. It is straightforward to extend [the method] to general count data.” However, they make no further mention of this extension, so applying our proposed method to that problem would be an obvious next step to build on this popular paper.
>
> However, the value of our problem solution to the genomics example should not be underestimated, although gene transcript count data is not as common a modality in machine learning. These data can be thought of as representing a snapshot image of the internal state the biological processes in a cell. Significant experimental resources are being used to collect this form of data to address many important biology and health questions. Despite the sparsity of the gene transcript counts in a cell screening, which is primarily due to the sampling process, is for the most part taken for granted. This likely hurts downstream analysis and the scientific interpretation of these experiments in hundreds of research labs worldwide. The single-cell genomics field is relatively new, but the amount of data has been growing exponentially, to the point where there have been recent attempts to train LLM foundation models using it (see scGPT [1], scFoundation [2]). We expect that it will become a more common data modality in machine learning literature in the near future.
>
> > I believe the method is also applicable to the non-central Hypergeometric as in the cited Sutter et al. (2022) paper and it would have been interesting to apply the proposed method to scenarios that are similar to those described in that work.
>
> We were not able to compare against the experiments in Sutter et al. (2022) because the starting assumptions about which parameters in the model are known/unknown are quite different. These authors address the non-central hypergeometric distribution, which is an extension of the hypergeometric distribution where the items belonging to each category have different probabilities of being sampled. Using the urn example, this would occur if different colored balls had different sizes, leading to some colors being more or less likely to be drawn from the urn, all else being equal. The standard hypergeometric distribution can be seen as a special case of the non-central distribution with all category weights equal. Therefore they learn the category weights while assuming all category sizes are known, whereas we learn the category sizes, while fixing the category weights to be equal. The experiments they perform primarily evaluate these learned category weights, which are constant in our method.

---

> ### Author Response · Authors · 2023-11-18
> **Author comment (2/2)**
>
> > It feels like there is a bit of a gap in the experiments, there is a jump from using 3 categories to more than 1000 categories. What about more intermediate cases, say with 100 categories? How does the proposed method work in this case with respect to n_max as well as the number of samples?
>
> During initial empirical exploration we found that the parameter inference worked well with an intermediate number of categories. We chose to use 1000 features for the experiment in Section 5 because in Section 6 we use 10,000 features, so across Sections 4-6 we span five orders of magnitude in terms of number of parameters to estimate. However, we agree that including an experiment comparing the effect of number of categories for Section 5 would strengthen the argument, and we are working to generate these results to add to the Appendix if time permits. We have added additional results for different f_max for the case of K=3 to the Appendix.
>
> We have performed more targeted investigation into the effects of n_max and f_max. One factor hindering this is the effect of the level of correlation between features. Our experiments on empirically simulated data, where we assumed no correlation structure between features, suggest that estimation performance degrades below an f_max (max counts sampled vs total number of counts) of roughly 0.25, as shown in Figure 2. However, we found that for the single-cell genomics data described in Section 6, the method can still perform well with a lower f_max. We believe this is due to the presence of mutual correlation among count features, in this case because genes are often turned on in groups, and act in biological pathways. Because in our approach we do not factorize the hypergeometric distribution, its parameters are inferred jointly, and therefore the joint correlation between features is beneficial to the estimation process. We expect this correlation is common in most real dataset, but its effect remains to be confirmed.
>
> > Has a similar relaxation (replacing a Binomial coefficient or factorials with a Gamma function) been used elsewhere that you are aware of in estimation contexts?
>
> The same relaxation that we use in our approach was also used for the non-central hypergeometric distribution by Sutter et al. (2022). The Gamma relaxation has been used to extend the binomial coefficient to negative values [4]. We are also aware that the Gamma function is typically used to compute binomial coefficients efficiently, such as in scipy. All sources we have found directly state that this switch from discrete to continuous is valid since x! = Gamma(x+1), by definition of the Gamma function.
>
> > There is a typo in equation (1), you should have c_2 and not n_2?
>
> Yes, you are correct, we have fixed that typo - thank you.
>
> > In equation (7), shouldn't you define c_i to be the largest category count over the T samples, as that gives you the lower bound on the N_k?
>
> You raise a good point. In the case of a single underlying distribution your suggestion would make more sense, but in the later scenarios we address, where there are multiple distributions to be learned, taking the max would not work as the samples are unlabelled (we don’t know which distribution they are drawn from), so we cannot compute the max within each category. We have added a note to make this distinction.
>
> ==References==
>
> [1] Cui, Haotian, et al. "scgpt: Towards building a foundation model for single-cell multi-omics using generative ai." bioRxiv (2023): 2023-04.
>
> [2] Hao, Minsheng, et al. "Large Scale Foundation Model on Single-cell Transcriptomics." bioRxiv (2023): 2023-05.
>
> [3] Thomas M. Sutter, Laura Manduchi, Alain Ryser, and Julia E. Vogt. Learning Group Importance using the Differentiable Hypergeometric Distribution (2022)
>
> [4] Kronenburg, M. J. "The binomial coefficient for negative arguments." arXiv preprint arXiv:1105.3689 (2011).

---

> > ### Comment · Reviewer_XXPF · 2023-11-22
> > **Response**
> >
> > Thank you for the authors' response. I have read it, and I will retain my original score. In my view, it would be good to say more about applications in a statistical, even if not a machine learning context, in order for this to be a more convincing paper.

---

### Official Review · Reviewer_gy2Q · 2023-11-07

**Soundness:** 2 fair
**Presentation:** 3 good
**Contribution:** 2 fair
**Rating:** 5
**Confidence:** 2

**Summary:**

The paper proposes to use continuous relaxations of hypergeometric likelihood, essentially the gamma functions instead of combinatorial seelctions. An application to sparse count data is shown.

**Strengths:**

The paper is largely well-written and the application of sparse single-cell genomics could be interesting especially since the proposed method does recover the true number of transcripts.

**Weaknesses:**

The paper is incomplete. The proposed method is very simple, hyper-geometric likelihoods are relaxed to continuous functions using gamma functions which is known from before. A penalty is added to the optimization problem to ensure that the total count always exceeds individual draws. There are no theoretical justifications on how far or close the proposed relaxations could be from the true count-based discrete distributions. There are only simulated experiments to validate the method. The application of single-cell genomics using VAEs is interesting but is not compared against any other baselines. The paper is probably better suited to a genomics conferences as that community would probably appreciate this a lot more.

**Questions:**

- Can there be theoretical arguments made on how bad/good the proposed relaxation would be ?

- Is there a relationship of the proposed method to other hyper-parameter optimization methods ? Why were no other baselines included that could possibly be used to solve the transcript problem? There is a huge literature on trying to estimate the hyper-parameters of count distributions  including the buffet processes.

- How exactly the fact about with/without replacement helpful for the genome problem? The dataset has counts high enough that even we use the methods that sample with replacement, they should possibly give reasonable results. Are these approaches known? if not, would it not make sense to compare against such baselines ?

- why is N>>1 required for the continuous relaxation to be reasonable ?

---

> ### Author Response · Authors · 2023-11-18
> **Author comment (1/2)**
>
> Thank you for your review! We have addressed your questions and comments below, and uploaded a new revision of the paper, implementing your suggestions and updating the figures.
>
> > The paper is incomplete. The proposed method is very simple, hyper-geometric likelihoods are relaxed to continuous functions using gamma functions which is known from before.
>
> We agree that the continuous relaxation of the hypergeometric distribution is not in itself a novel contribution, as there are a few instances of it being used previously. However, we are convinced that the primary contributions are the demonstration that estimation of unknown ground-truth population sizes is tractable, which, to the best of our knowledge, has not been previously shown, as well as the extension to more complex data generating distributions.
>
> In fact, part of the strength of this method is its simplicity and we hope to encourage further attempts to directly model count distributions, as opposed to their relative abundances. It is surprising to us that this approach has not yet been proposed, and this paper hopes to fill that gap.
>
> The multinomial model is one of the most commonly used distributions for probabilistic models, and the hypergeometric distribution is its analog when it is necessary to account for the dependence between observations. In other words, moving from an assumption of i.i.d sampling to an assumption of exchangeable sampling leads to the switch from multinomial to hypergeometric model. Having said this, there have been very few recent methods that leverage the hypergeometric distribution, but we believe this is because there are few modern count datasets where the true number of counts is known.
>
> As an example of the current interest for modeling counts, in the highly cited Liang et al. (2018) paper, the main contribution is to model the user-by-item click (count) matrix using a VAE with a multinomial likelihood. They state that “for simplicity, we binarize the click matrix. It is straightforward to extend [our method] to general count data.” However, this extension is never mentioned again in that work. We believe we are the first to directly address this extension and show that it is tractable.
>
> > Can there be theoretical arguments made on how bad/good the proposed relaxation would be?
>
> At a minimum we can say that the continuous relaxation of the hypergeometric distribution will be exact for integer values of x, given that x! = Gamma(x+1).
>
> In the only recent use of a continuous relaxation of the hypergeometric distribution that we could find, Sutter et al. (2022) use a Kolmogorov-Smirnov test to show that their proposed continuous relaxation of the non-central hypergeometric distribution is statistically indistinguishable in sample distribution from the discrete form. The authors similarly replaced the factorial with the Gamma function, although they factor the joint multivariate distribution into bivariate conditional distributions (see Section 4 of their paper). Because we do not need to factorize the joint distribution, our proposed relaxation should perform at least as well, and therefore we can conclude that our proposed relaxation is also identical in distribution to the discrete hypergeometric distribution.

---

> ### Author Response · Authors · 2023-11-18
> **Author comment (2/2)**
>
> > Is there a relationship of the proposed method to other hyper-parameter optimization methods ? Why were no other baselines included that could possibly be used to solve the transcript problem? There is a huge literature on trying to estimate the hyper-parameters of count distributions including the buffet processes.
>
> We agree that there is a large literature on trying to estimate the hyperparameters of count distributions, and recently transcript count distributions in particular. However, the majority of these previous approaches attempt to learn the distribution of the observed (undersampled) counts, whereas here we aim to obtain the true distribution of counts before sampling. In other words, most methods attempt to estimate the parameters of distributions that account for the sparsity of the data (such as with a (zero inflated) binomial model), whereas we assume the sparsity is due to undersampling and try to estimate the parameters of the original distribution prior to sampling. Because we attempt to directly model the true underlying distribution, we are not able to provide a comparison.
>
> As far as we are aware, no existing method has attempted to estimate this true underlying count distribution in the population under the assumption of sampling without replacement, which is why no baseline could be provided.
>
> > How exactly is the fact about with/without replacement helpful for the genome problem? The dataset has counts high enough that even if we use the methods that sample with replacement, they should possibly give reasonable results. Are these approaches known? If not, would it not make sense to compare against such baselines ?
>
> We agree that the assumption that transcript counts are sampled with replacement (eg. a multinomial likelihood) may be reasonable for this problem if we are only interested in learning a useful latent space. However, there are two limitations in this scenario. The first and most important is that the assumption of sampling with replacement would only permit the model to learn the relative abundance of different feature counts (genes) through a probability distribution over the features. However, we believe that knowing the true count of RNA in a cell is valuable for downstream bioinformatic analyses. At a minimum, it significantly reduces sampling noise, which would improve the performance of further in-depth analyses. In the other applications we mention, such as recommender systems, we similarly argue that in some use cases the true number of counts is valuable. Second, gene counts have a high dynamic range, with some genes have many counts and some having very few, spanning 3-4 orders of magnitude. Therefore, while the approximation resulting from the assumption of sampling with replacement may work well for modeling genes with high counts, it introduces more significant errors for genes with low counts, where the probability of sampling a second transcript changes significantly after a first has been sampled. In fact, it is known that most signal in single-cell data is driven by the high abundance genes [3], which may be due to this bias.
>
> > Why is N>>1 required for the continuous relaxation to be reasonable ?
>
> We added this caveat specifically when thinking about the maximum percent error introduced by approximating an integer with a whole number, but in fact N>>1 is not required. We have removed that sentence for the sake of clarity.
>
> ==References==
>
> [1] Liang, Dawen, et al. "Variational autoencoders for collaborative filtering." Proceedings of the 2018 world wide web conference. 2018.
>
> [2] Thomas M. Sutter, Laura Manduchi, Alain Ryser, and Julia E. Vogt. Learning Group Importance using the Differentiable Hypergeometric Distribution (2022)
>
> [3] Squair, J.W., Gautier, M., Kathe, C. et al. Confronting false discoveries in single-cell differential expression. Nat Commun 12, 5692 (2021).

---

> > ### Comment · Reviewer_gy2Q · 2023-11-22
> >
> > Thank you for your response.
> >
> > After going through your response and the other reviews, I am sorry but the paper falls below par. Here are some suggestions to improve -- (1) Add more applications that are not just bioinformatics. If you would like to stick to bioinformatics, may be a conference/journal that can appreciate its importance is more useful. (2) Add more baselines in experiments. I understand that the other methods do not estimate true underlying counts but that should help your method perform better in the experiments and practically illustrate where the existing methods fall short. (3) Add more theoretical/intuitive justifications by expanding on your response.

---

### Author Response · Authors · 2023-11-21

We hope the Reviewers have had a chance to read our responses to their comments. We would like to remind them that the end of the discussion period is approaching.

---

### Meta-Review · Area_Chair_Zg7i · 2023-12-05

**Metareview:**

This paper tackles the problem of maximum likelihood estimation in a multivariate hypergeometric model, where both the size of the total population and its constituents categories are unknown. The proposed approach relies on a continuous relaxation. While the approach is sound and the paper clearly written, the reviewers noted a poor fit for ICLR due to the limited experiments and lack of suitable ML baselines. Some concerns were also raised by a reviewer regarding the motivation of the approach, which remained even after the author's response.   For these reasons, I do not think this article is suitable for ICLR, and recommend a rejection.

**Justification For Why Not Higher Score:**

Weak experimental part - lack of more extensive ML baselines - remaining questions regarding the overall motivation

**Justification For Why Not Lower Score:**

N/A

---

### Decision · Program_Chairs · 2024-01-16

Reject